# Highly Sensitive and Transparent Urea-EnFET Based Point-of-Care Diagnostic Test Sensor with a Triple-Gate a-IGZO TFT

**DOI:** 10.3390/s21144748

**Published:** 2021-07-12

**Authors:** Seong-Kun Cho, Won-Ju Cho

**Affiliations:** Department of Electronic Materials Engineering, Kwangwoon University, 20, Gwangun-ro, Nowon-gu, Seoul 01897, Korea; whtjdrms98@gmail.com

**Keywords:** ion-sensitive field-effect transistor, triple-gate, extended gate, capacitive coupling, urea sensor, antigen-antibody

## Abstract

In this study, we propose a highly sensitive transparent urea enzymatic field-effect transistor (EnFET) point-of-care (POC) diagnostic test sensor using a triple-gate amorphous indium gallium zinc oxide (a-IGZO) thin-film pH ion-sensitive field-effect transistor (ISFET). The EnFET sensor consists of a urease-immobilized tin-dioxide (SnO_2_) sensing membrane extended gate (EG) and an a-IGZO thin film transistor (TFT), which acts as the detector and transducer, respectively. To enhance the urea sensitivity, we designed a triple-gate a-IGZO TFT transducer with a top gate (TG) at the top of the channel, a bottom gate (BG) at the bottom of the channel, and a side gate (SG) on the side of the channel. By using capacitive coupling between these gates, an extremely high urea sensitivity of 3632.1 mV/pUrea was accomplished in the range of pUrea 2 to 3.5; this is 50 times greater than the sensitivities observed in prior works. High urea sensitivity and reliability were even obtained in the low pUrea (0.5 to 2) and high pUrea (3.5 to 5) ranges. The proposed urea-EnFET sensor with a triple-gate a-IGZO TFT is therefore expected to be useful for POC diagnostic tests that require high sensitivity and high reliability.

## 1. Introduction

Urea sensors are utilized in various fields, such as food [1,2], environmental protection [3,4], pharmaceuticals, fertilizers, and especially bio-clinical analysis [5]. In the human body, urea is the main end-product of nitrogen metabolism, and it is mainly removed from the body in urine but is also secreted in body fluids such as blood, sweat, and saliva. Therefore, we can diagnose and/or treat pathologies such as renal insufficiency, hyperpyrexia, leukemia, diarrheal diseases, diabetes mellitus, and hyperthyroidism by identifying changes in the urea concentration of the blood, serum, or urine [6]. In general, the level of urea in serum ranges from 2.5 mM to 7.5 mM and rapidly increases under pathophysiological conditions, providing key information for the diagnosis of renal function and various kidney and liver diseases. Chronic kidney disease or end-stage renal disease is generally caused by a progressive loss of kidney function; this increases the level of urea in the blood, which is known as azotemia or uremia. At urea levels above 30 mM, renal insufficiency becomes important, and hemodialysis is required [7]. Therefore, it is crucial to develop new technologies to assess urea levels quickly and at a low cost.

The ion-sensitive field-effect transistor (ISFET), first introduced by Bergveld in the early 1970s [8], has been used extensively for physiological measurements in biomedical applications. Because ISFETs have a metal-oxide-semiconductor FET (MOSFET) device structure, they can be manufactured using the complementary metal-oxide-semiconductor (CMOS) process, which is the most promising technology for microsystems. Unlike MOSFETs, however, ISFETs do not have metal or polysilicon gate electrodes, which exposes the gate oxide insulator to the electrolyte solution. In an ISFET, the gate voltage is supplied to the reference electrode immersed in the same electrolyte solution, instead of to the gate electrode, as in a MOSFET. Thus, the gate potential, which modulates the drain current, is defined by the voltage supplied by the reference electrode and the surface potential of the sensing membrane. As indicated by the site binding model, the surface potential of the sensing membrane is determined by the ions adsorbed on the gate insulator surface of the inner Helmholtz plane in the Stern-Gouy-Chapman model of the electrical double layer [9,10]. Semiconductor-based ISFETs are compatible with the CMOS manufacturing processes, enabling mass production and miniaturization. Conversely, the gate insulators (i.e., sensing membranes) of an ISFET are susceptible to damage by the chemical ions in the electrolyte solution [11]. Therefore, an ISFET with an immobilized enzyme extended gate (EG) was proposed by Clark and Lyons and introduced as the urea-enzymatic FET (EnFET) [12,13]. In the EnFET, the detection of urea is accomplished through a bio-signal delivered to the pH-ISFET sensor platform via a reaction with urease enzymes immobilized on the surface of the EG sensing membrane. Figure 1 shows the sensing mechanism of the urease enzymes immobilized SnO_2_ sensing membrane. Since SnO_2_ sensing membrane without immobilized urease does not react with urea, the surface potential does not change. On the other hand, since the urease-immobilized SnO_2_ sensing membrane reacts with urea and forms a reaction product, the surface potential changes. Urea hydrolysis is catalyzed by urease, as described in the following expression:$${\left({\mathrm{NH}}_{2}\right)}_{2}\mathrm{CO}+3H_{2}O\overset{{}^{\circ}\mathrm{urease}{}^{\circ}}{\to }{\mathrm{CO}}_{2}+2{\mathrm{NH}}_{4}^{+}+2{\mathrm{OH}}^{-}.$$

The reaction product increases the pH at the surface of the sensing membrane. If the urease enzyme is not immobilized, there is no change in the surface potential because there is no reaction product. The potential change between the sensing membrane surface and the urea solution allows the pH-ISFET sensor platform to indirectly determine the urea concentration by calibrating the change in the major parameters of the EnFET in the sensing operation [13]. For this reason, the pH-ISFET sensor platform can be applied as a urea-EnFET. However, the sensitivity of the conventional pH-ISFET sensor platform has a Nernst limit (59.5 mV/pH at room temperature) that is not suitable for detecting weak biological signals. This poor sensitivity constrains the detection of biomaterials with small concentrations and low amplitude biological signals in the human body, such as DNA, cells, antigen antibodies, enzymes, and hormones [14,15,16,17].

In this study, we overcome the fundamental chemical damage problem of the gate insulator in an FET transducer by constructing a urea-EnFET sensor with an EG that is separate from the FET transducer. This configuration is expected to produce a cost-effective sensor device because the low-cost EG (disposable detector) can easily be exchanged, thereby allowing the continued use of expensive and sophisticated FET transducers without damage. To implement a transparent urea-EnFET sensor, we fabricated a transducer unit based on a triple-gate amorphous indium gallium zinc oxide (a-IGZO) thin film transistor (TFT) on a glass substrate. In addition, we fabricated a sensing unit by urease immobilization through surface treatment on SnO_2_ sensing membrane EG. We implemented the urea-EnFET sensor by electrically connecting triple-gate a-IGZO TFTs and urease-immobilized SnO_2_ sensing membrane EG. Therefore, we achieved increased sensitivity to weak signal biomaterials in the a-IGZO TFT design with a triple-gate structure that consists of a top gate (TG), bottom gate (BG), and side gate (SG) at the top, bottom, and side of the TFT channel, respectively. Capacitive coupling occurs between these triple gates, allowing significantly amplified sensitivity. By applying this unique pH-ISFET sensor platform to the urea-EnFET, we achieved a very high urea sensitivity of 3632.1 mV/pUrea in the range of pUrea 2 to 3.5, which is 50 times greater than that reported in previous studies. Moreover, we demonstrate that high urea sensitivity is reliably obtained, even in the low pUrea range (0.5 to 2) and the high pUrea range (3.5 to 5). Accordingly, the urea-EnFET sensor platform proposed in this paper is a useful portable sensing tool for urea detection, and it is expected to serve as a highly sensitive sensor platform for point-of-care (POC) diagnostic tests for the detection of a variety of biomaterials, such as DNA, cells, antigen antibodies, enzymes, and hormones.

## 2. Materials and Methods

### 2.1. Fabrication Process of the Triple-Gate a-IGZO TFT Transducer Unit

A 1 × 1 cm^2^ transparent glass substrate (Corning Inc., New York, NY, USA) was used as the substrate of the a-IGZO TFT. To fabricate the triple-gate a-IGZO TFT transducer, a 300-nm-thick indium tin oxide (ITO) layer and a 300-nm-thick SiO_2_ film were deposited sequentially as the BG and the bottom gate insulator, respectively, using an RF magnetron sputtering system. ITO was deposited at a deposition power of 100 W, a working pressure of 3 mTorr in an ambient of Ar with a flow rate of 20 sccm, and SiO_2_ was deposited at a deposition power of 200 W, a working pressure of 3 mTorr in an ambient of Ar/O_2_ with a flow rate of 30/2 sccm. Then, a 20-nm-thick a-IGZO (In:Ga:Zn = 1:1:1) film was deposited by the RF magnetron sputtering system at a deposition power of 100 W, a working pressure of 6 mTorr in an ambient of Ar with a flow rate of 30 sccm as the active region. The active channel region (width/length = 20/10 μm) was defined using photolithography and wet etching processes. To form the source/drain (S/D), a 100-nm-thick ITO layer was deposited by an RF magnetron sputtering system and a lift-off process. Subsequently, a 10/35-nm-thick SiO_2_/Ta_2_O_5_ stacked top gate insulator was deposited on the active layer using RF magnetron sputtering system. SiO_2_ was deposited at a deposition power of 200 W, a working pressure of 3 mTorr in an ambient of Ar/O_2_ with a flow rate of 30/2 sccm, and Ta_2_O_5_ was deposited at a deposition power of 75 W, a working pressure of 3 mTorr in an ambient of Ar with a flow rate of 20 sccm. The SiO_2_/Ta_2_O_5_ stacked top gate insulator increases the capacitive coupling with the capacitance of the bottom gate oxide and reduces the gate leakage current. After the deposition of a 150-nm-thick ITO film by RF magnetron sputtering, the TG and SG were formed simultaneously using photolithography and lift-off processes. Finally, to improve the contact characteristics of the gate electrodes, post-deposition annealing was performed at 250 °C for 30 min in O_2_ ambient using a resistive heating furnace. A schematic diagram of the fabricated triple-gate a-IGZO TFT is shown in Figure 2a, and optical microscopy and photographic images of the device are shown in Figure 2b and Figure 2c, respectively.

### 2.2. Fabrication Process of the Urease-Immobilized SnO_2_ Sensing Membrane EG Sensing Unit

To fabricate the urease-immobilized SnO_2_ sensing membrane EG sensing unit, a 300-nm-thick ITO electrode layer and a 50-nm-thick SnO_2_ sensing membrane were sequentially deposited on a 1.5 × 2.5 cm^2^ transparent glass substrate (Corning Inc., New York, NY, USA) using RF magnetron sputtering system. SnO_2_ was deposited at a deposition power of 100 W, a working pressure of 3 mTorr in an ambient of Ar with a flow rate of 30 sccm. Then, covalent bonding was used to immobilize urease on the SnO_2_ sensing membrane of the EG. Figure 3a is a schematic of the covalent bonding process used to immobilize urease enzymes on the SnO_2_ sensing membrane of the EG. The covalent bonding process consists of four steps: (1) the SnO_2_ surface of the EG was treated with O_2_ plasma to generate hydroxyl groups; (2) the EGs were silylated by immersion in a 40 °C, 9% 3-aminopropyltriethoxysilane (APTS) solution for 4 h; (3) the samples were placed in 10% glutaraldehyde (GA), a dual function group for crosslinking urease amines and APTS, for 1 h; and (4) the urease powder was mixed with the phosphate buffer solution (PBS) at a concentration of 1.5 μg/mL, and the sample surfaces were soaked in the urease solution. The EG specimens were then stored overnight in a refrigerator at 4 °C to immobilize the urease enzymes, after which the immobilized enzyme was rinsed with PBS. After being rinsed with PBS, a 0.6 cm diameter polydimethylsiloxane (PDMS) reservoir for urea electrolyte storage was fixed on the surface-treated SnO_2_. All the reagents were purchased from Sigma Aldrich Co. (St. Louis, MO, USA) and used without further purification. The urease-immobilized SnO_2_ sensing membrane EG is connected to the dark box through an electrical cable, which is connected to the top gate of the triple-gate a-IGZO TFT through a positioner. Figure 2a is a schematic diagram showing the electrical connection between the fabricated urease-immobilized SnO_2_ sensing membrane EG and triple-gate a-IGZO TFTs, and a photographic image of the fabricated EG is shown in Figure 2d.

To verify the immobilization of the urease enzymes, the surface morphology of the SnO_2_ membranes was analyzed. Atomic force microscopy (AFM; SPM Solver-Pro, NT-MDT, Moscow, Russia) and scanning electron microscopy (SEM; Sirion 400, FEI Company, Hillsboro, OR, USA) images of the SnO_2_ membrane surfaces after the GA procedure and urease enzyme immobilization are shown in Figure 3b,c. After the third surface treatment, the GA layer undergoes automatic polymerization on the SnO_2_ surface and forms large domains. The aldehyde groups in GA can be used to bind urease enzymes covalently through the exposed amine groups of lysine amino acids. SEM images taken after urease enzyme immobilization confirm the presence of crystal structures over and throughout the SnO_2_ membrane surface, and AFM images show that the root-mean-square roughness (R_q_) of the surface increased significantly from 0.5804 nm to 3.411 nm.

### 2.3. Device Characterization

The current-voltage characteristics of the triple-gate a-IGZO TFT-based ISFET sensor platform were measured using an Agilent 4156B precision semiconductor parameter analyzer. The Ag/AgCl reference electrode (Horiba 2080A-06T) was used for pH and urea sensing. All measurements were conducted in a dark box to avoid interference from external light and noise. We evaluated the pH sensitivity in the TG, BG, and SG modes. We then examined the reliability and stability by measuring the hysteresis and drift effects operating between the pH electrolytes and the sensing membrane. After evaluating the pH sensing performance of the triple-gate a-IGZO TFT-based ISFET sensor platform, the urea sensitivity was evaluated in the TG, BG and SG modes.

The proposed triple-gate a-IGZO TFT-based ISFET sensor platform is capable of detection in three operating modes, which are shown in Figure 4: (a) TG-biasing TG-sensing (TG mode), (b) BG-biasing TG-sensing (BG mode), and (c) SG-biasing TG-sensing (SG mode). The TG mode functions like a conventional ISFET, where the BG electrode is grounded and measures the drain current while sweeping the TG bias voltage is delivered through the reference electrode, as shown in Figure 4a. In contrast, the BG mode measures the drain current while grounding the TG electrode and sweeping the BG bias voltage, as shown in Figure 4b. The SG mode measures the drain current while sweeping the SG bias voltage by grounding the TG and floating the BG, as shown in Figure 4c. In particular, BFG in Figure 4c refers to the BG in a floating state. Table 1 summarizes the bias conditions for each gate voltage in the three operating modes of the triple-gate a-IGZO TFT-based ISFET sensor platform.

Figure 5 shows (a) a cross-sectional schematic and (b) an equivalent circuit of a triple-gate a-IGZO TFT-based ISFET sensor platform. The black, red, and blue dotted boxes in Figure 5 correspond to the TG, BG, and SG modes of the ISFET sensor, respectively.

In general, conventional ISFETs driven by the TG mode have a sensitivity limit (ideal Nernst response at 300 K of 59.5 mV/pH) according to the site binding model [18]. This is because the threshold voltage of the TG ($V_{TG}^{th}$) is driven by changes in the TG mode, in accordance with the surface potential (ψ) of the sensing membrane from the electrolyte:(1)$$\Delta V_{th}^{TG}=-\Delta \psi$$
where $\Delta V_{th}^{TG}$ is the threshold voltage shift of the TG.

In contrast, in the BG mode, the sensitivity can be amplified by the capacitive coupling effect between the top and bottom of the TFT channel. Thus, the threshold voltage of the BG ($V_{th}^{BG}$) driven by the BG mode is given not only by ψ from electrolyte but also by the capacitance ratio between $C_{upper}$ and $C_{lower}$. In this mode, $C_{upper}$ is a series combination of the TG capacitance $C_{tox}$ and the channel depletion capacitance $C_{IGZO}$, while $C_{lower}$ is the BG capacitance $C_{box}$. Then, the sensitivity of the BG mode according to the BG voltage sweep is given by Equation (2) [19]:(2)$$\Delta V_{th}^{BG}=-\frac{C_{upper}}{C_{lower}}\Delta \psi =\frac{C_{IGZO}C_{tox}}{C_{box}\left(C_{IGZO}+C_{tox}\right)}\Delta V_{th}^{TG},$$
where $\Delta V_{th}^{BG}$ is the threshold voltage shift of the BG. This means that the ψ of the sensing membrane is transferred to the TG and results in $\Delta V_{th}^{BG}$ being shifted in proportion to $C_{tox}/C_{box}$. Thus, for the BG mode, a thicker BG oxide and thinner TG oxide are more desirable to further enhance the sensitivity by increasing the capacitive coupling ratio. This also implies that TG insulator engineering using high-k materials is effective for increasing $C_{upper}$ while reducing the leakage current, thereby improving the sensitivity. Nevertheless, increasing the thickness of the BG oxide entails a longer manufacturing process time and higher cost, while thinner TG oxides cause higher leakage currents, making this an inefficient approach. In the device proposed in this study, $C_{IGZO}$, $C_{tox}$, and $C_{box}$ are about 1.416 pF, 0.49 pF, and 0.023 pF, respectively, and substituting them into Equation (2), the theoretical amplification ratio of BG mode is about 16.

Meanwhile, the proposed triple-gate a-IGZO TFT-based ISFET sensor platform has an excellent SG mode for sensitivity amplification. We introduced the SG mode to increase the amplification ratio by further reducing the $C_{lower}$. In this case, the $C_{lower}$ of the device is $\left(C_{box}\times C_{sox}\right)/\left(C_{box}+C_{sox}\right)$ because the floated BG connects $C_{box}$ and $C_{sox}$ in series. $C_{sox}$ is the capacitance between the SG and BG and results in the SG gate effectively doubling the thickness of the BG oxide [20]. Therefore, since $C_{lower}$ is smaller in SG mode than in BG mode, SG mode is amplified more than BG mode. For the SG mode using an SG voltage sweep, the capacitive coupling and sensitivity are described in Equation (3) [21]:(3)$$\Delta V_{th}^{SG}=-\frac{C_{upper}}{C_{lower}}\Delta \psi =\frac{C_{IGZO}C_{tox}\left(C_{box}+C_{sox}\right)}{\left(C_{IGZO}+C_{tox}\right)\left(C_{box}C_{sox}\right)}\Delta V_{th}^{TG},$$
where $\Delta V_{th}^{SG}$ is the threshold voltage shift of the SG. In this mode, $C_{upper}$ is a series combination of $C_{tox}$ and $C_{IGZO}$, and $C_{lower}$ is a series combination of $C_{box}$ and $C_{sox}$. In the device proposed in this study, $C_{sox}$ is about 0.015 pF, and the theoretical amplification ratio of SG mode is about 40. This suggests that the sensitivity can be further amplified by using the SG mode of triple-gate a-IGZO TFTs.

## 3. Results

### 3.1. Electrical Characteristics of the Triple-Gate a-IGZO TFT

Figure 6 shows the typical electrical characteristics of triple-gate a-IGZO TFTs operating under TG, BG, and SG bias voltage sweep conditions corresponding to the TG, BG, and SG modes of the ISFETs, respectively. The transfer characteristic curves were measured at a drain voltage of 1 V, and the gate voltage was swept from −0.5 to 1.0 V for TG mode, from −4 to 10 V for BG mode, and from −10 to 25 V using the double-sweep mode to evaluate hysteresis characteristics. Meanwhile, the output characteristics were measured from V_TG_ − V_TH_ = 0~0.5 V in 0.05 V steps for the TG mode, V_BG_ − V_TH_ = 0~5 V in 0.5 V steps for the BG mode, and V_SG_ − V_TH_ = 0~15 V in 1.5 V steps for the SG mode. In the transfer characteristic curves and the output characteristic curves of each mode, it can be seen that the drain current is adjusted according to the drain voltage and the gate voltage, respectively.

Table 2 summarizes the electrical parameters extracted from the transfer characteristic curves. A threshold voltage (V_TH_) of −0.11 V, hysteresis voltage (V_HYS_) of 0.02 V, field-effect mobility (μ_FE_) of 7.37 cm^2^/V·s, subthreshold swing (SS) of 157.07 mV/dec, and on/off current ratio (I_ON_/I_OFF_) of 6.78 × 10^7^ were obtained under top gate operation. Meanwhile, V_TH_, V_HYS_, μ_FE_, SS, and I_ON_/I_OFF_ of 0.13 V, 0.12 V, 12.89 cm^2^/V·s, 356.90 mV/dec, and 5.62 × 10^7^ were obtained under bottom gate operation, and V_TH_, V_HYS_, μ_FE_, SS and I_ON_/I_OFF_ of −0.28 V, 0.27 V, 19.38 cm^2^/V·s, 434.13 mV/dec, and 4.48 × 10^7^ were obtained under side gate operation. The TG, BG, and SG sweep operations of the fabricated triple-gate a-IGZO TFT all show excellent electrical characteristics, meaning that they are satisfactory as sensors for sensing pH or urea.

### 3.2. pH Sensing Characteristics of the Triple-Gate a-IGZO TFT-Based pH-ISFET Sensor Platform

Figure 7 shows the transfer characteristic curves and pH sensitivities of the triple-gate a-IGZO TFT-based pH-ISFET sensor platform for various pH concentrations in the TG, BG, and SG modes. The measurements were taken using the EG with an SnO_2_ membrane without immobilized urease. The reference voltage for determining the shift of the threshold voltage with respect to the pH concentration was defined as each gate voltage at the reference drain current of 1 nA (I_ref_). From the transfer characteristic curves, it is evident that reference voltage increases for pH concentrations ranging from pH 3.07 to 9.87. The increase in the total reference voltage in this pH range is 0.39 V for the TG mode, 7.14 V for the BG mode, and 16.23 V for the SG mode, while the pH sensitivity, which corresponds to the slope of the linear fits, is 55.58, 1038.89, and 2363.9 mV/pH for the TG, BG, and SG modes, respectively. These results show that compared to the TG mode, the BG mode is amplified by 18.7 times and the SG mode by 42.5 times, demonstrating that the proposed sensor platform can greatly amplify the pH sensitivity according to the sensing mode, as shown in Equations (2) and (3).

Figure 8 shows the stability and reliability test of the triple-gate a-IGZO TFT-based pH-ISFET sensor platform as determined by measuring the hysteresis voltage and drift rate. The hysteresis voltage shown in Figure 8a is caused by the micro potential charge of the sensing membrane, which occurs when the ions in the pH electrolyte react slowly with the sensing membrane [22]. We defined the hysteresis voltage as the difference between the reference voltage for the first and last pH 7 in the pH loop (pH 7 → 10 → 7 → 4 → 7). The hysteresis voltages for the TG, BG, and SG modes were 12.01, 74.30, and 115.10 mV, respectively. The drift rate is the change in reference voltage per hour, which results from the micro potential charge caused when ions penetrate the sensing membrane over a long period [23]. The drift rates in the TG, BG, and SG modes were 13.89, 60.39, and 109.36 mV/h, respectively, as shown in Figure 8b. The sensing parameters of triple-gate a-IGZO TFT-based pH-ISFET sensor platform are summarized in Table 3. Compared to the TG mode, the BG mode has 18.7 times higher sensitivity, 6.2 times greater hysteresis voltage, and 4.3 times higher drift rate, while the SG mode has 42.5 times higher sensitivity, 9.6 times larger hysteresis voltage, and 7.9 times higher drift rate. This suggests that the SG mode has better stability and reliability than the other modes because its increases in hysteresis, voltage and drift rates are relatively low compared to its increase in sensitivity.

### 3.3. Urea Sensing Characteristics of the Triple-Gate a-IGZO TFT-Based Urea-EnFET Sensor Platform

After immobilization of the urease enzyme, we performed a control test to assess the selectivity of the urea-EnFET sensor platform and to verify whether it selectively detects the urease-urea interaction. In the case of the SnO_2_-urea interaction, there is no shift in the reference voltage, as shown in Figure 9a, but for the urease-urea interaction, a shift in reference voltage can be observed in Figure 9b. This indicates that the urease enzyme was immobilized on the SnO_2_ and that it reacted with urea, verifying the selectivity. Furthermore, to ensure the reliability and stability of the measurements, we measured the variation in the drain current over time after supplying various concentrations of urea electrolytes to the EG reservoir and measured the drain current-time in the SG mode while biasing a constant voltage (V_SG_: 15 V, V_D_: 1 V) for 300 s. Figure 9c shows the measured response time of the urea-EnFET sensor platform for various concentrations of urea electrolytes ranging from pUrea 0.5 to pUrea 5. The urease-urea interaction occurred for 120 s and reached equilibrium through diffusion of the end-product. This suggests that the proposed urea-EnFET sensor platform can reliably detect urea electrolyte in about 300 s after the urease-urea reaction is initiated. We therefore measured the transfer characteristic curves after 300 s of supplying urea electrolyte to the EG reservoir for stable measurement.

Figure 10 shows the sensing characteristics for the urease-urea reaction of the triple-gate a-IGZO TFT-based urea-EnFET sensor platform in the TG, BG, and SG modes for various urea concentrations. To validate the triple-gate a-IGZO TFT-based urea-EnFET sensor platform, we detected urea for a wide range of urea concentrations: from pUrea 0.5 to pUrea 5 (10 μM to 316 mM) in PBS buffer solution. The transfer characteristic curves for the urea-EnFET sensor platform in the TG, BG, and SG modes as a function of urea concentration are shown in Figure 10a,b, and Figure 10c, respectively. The end-product of the urease–urea reaction decreases the surface potential of the SnO_2_ sensing membrane such that when the urea concentration increases, the surface potential decreases, and the reference voltage shifts in a positive direction. As a result, the shift of the transfer characteristic curve was larger in the BG mode than in the TG mode and greater in the SG mode than in the BG mode. Figure 10d shows the shift in the reference voltage in the TG, BG, and SG modes for various urea concentrations, as extracted from Figure 10a–c. The shift in the reference voltage due to the urease-urea reaction can be divided into three regions: (R1) a small change in the reference voltage for large urea concentrations from pUrea 0.5 to pUrea 2, (R2) the greatest change in the reference voltage from pUrea 2 to pUrea 3.5, and (R3) a small change in the reference voltage for low urea concentrations from pUrea 3.5 to pUrea 5. The low response in R1 is due to the excessive concentrations of OH− ions and by-products of the urease and urea reaction. At high OH^−^ ion concentrations in the electrolytes, the urease enzyme exhibits lower reactivity toward urea. The high response in R2 is most useful for urea electrolyte detection because it results in a higher voltage shift, whereas the low voltage shift in R3 is due to the low urea concentration.

Figure 11 shows the urea sensitivity for the SG, BG, and TG modes of the triple-gate a-IGZO TFT-based urea-EnFET sensor platform in the R1, R2, and R3 regions extracted from Figure 10d. In R1, the urea sensitivity in the TG, BG, and SG modes was 33.2, 436.3, and 1272.7 mV/pUrea, respectively. In R2, where the largest signal change was observed, the urea sensitivity in the TG, BG, and SG modes was 90.1, 1279.0, and 3632.1 mV/pUrea, respectively. Finally, in R3, the urea sensitivity of the TG, BG, and SG modes was reduced to 45.3, 441.5, and 1051.0 mV/pUrea, respectively. This result suggests that the urea sensitivity of the proposed urea-EnFET sensor platform can be amplified in the BG and SG modes similarly to the sensitivity amplification of the pH-ISFET sensor platform. We found that the SG mode is suitable for detecting urea electrolytes at a wide range of concentrations. Table 4 summarizes the characteristics of previously reported urea sensors for comparison with the proposed urea sensor. The urea-EnFET sensor fabricated in this study exhibits properties superior to those of the previously reported urea sensors, in particular, its exceptionally high sensitivity and linearity for a wide range of urea concentrations from pUrea 0.5 to 5.

## 4. Conclusions

In this study, we investigated a highly sensitive, transparent urea-EnFET based POC sensor using a triple-gate structure a-IGZO TFT. The urea-EnFET sensor consists of an SnO_2_ EG with immobilized urease for the sensing unit and a triple-gate a-IGZO TFT for the transducer unit. The proposed triple-gate a-IGZO TFT transducer has a TG at the top of the channel, a BG at the bottom of the channel, and a SG on the side of the channel, and it exhibits excellent sensitivity amplification for urea detection. In particular, in the SG mode, we were able to detect urea with a very high sensitivity of 3632.1 mV/pUrea in the pUrea range of 2 to 3.5; this is about 50 times larger than the sensitivity reported in previous works. Furthermore, high urea sensitivity and reliability were even obtained in the low pUrea range of 0.5 to 2 and high pUrea range of 3.5 to 5. We therefore expect the proposed urea-EnFET sensor with a triple-gate a-IGZO TFT and SnO_2_ EG with immobilized urease to be a useful platform for POC diagnostic tests that require high sensitivity and high reliability.

## Figures and Tables

**Figure 1 sensors-21-04748-f001:**
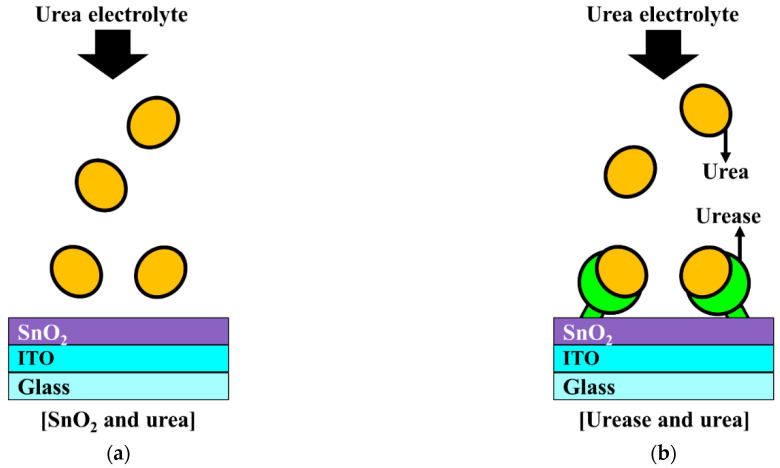
Schematic illustration of (**a**) the interaction between the SnO_2_ membrane and urea and (**b**) the interaction between urease and urea.

**Figure 2 sensors-21-04748-f002:**
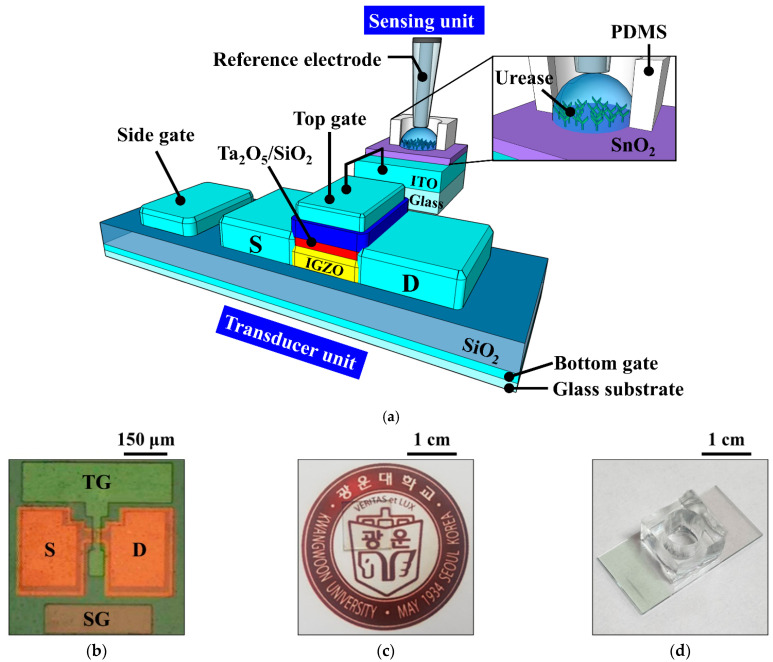
(**a**) Schematic diagram of the urea-EnFET sensor configured by the triple-gate a-IGZO TFT and urease-immobilized SnO_2_ sensing membrane EG; (**b**) optical microscopy and (**c**) photographic images of the fabricated triple-gate a-IGZO TFT; and (**d**) photographic image of the urease-immobilized SnO_2_ sensing membrane EG.

**Figure 3 sensors-21-04748-f003:**
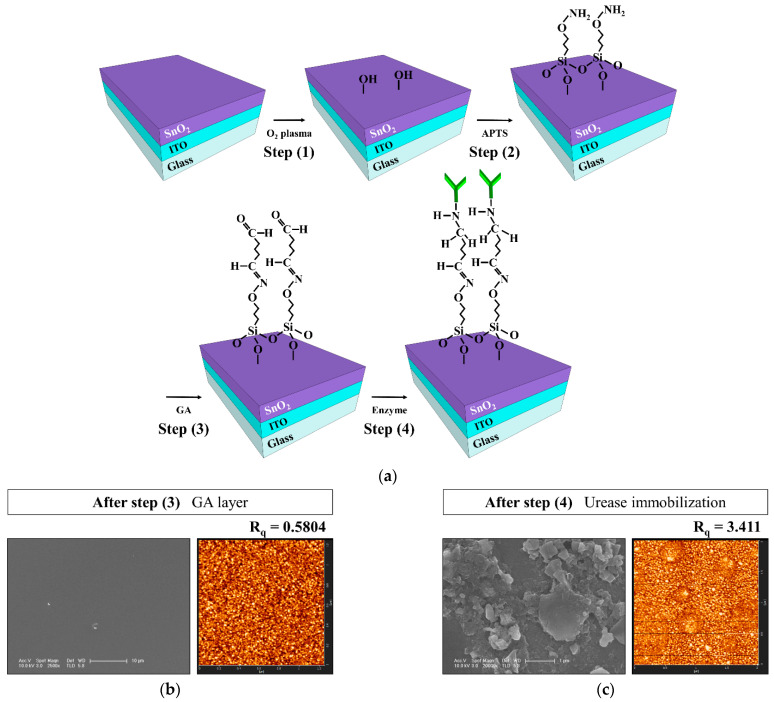
(**a**) Schematic diagram of the covalent bond process on the SnO_2_ sensing membrane of the EG. SEM and AFM images of the SnO_2_ membrane surface after (**b**) Step 3 (GA) and (**c**) Step 4 (immobilization of the urease enzymes).

**Figure 4 sensors-21-04748-f004:**
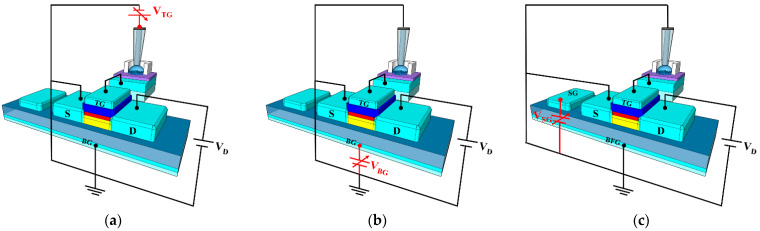
Schematics of the three different sensing modes of the proposed triple-gate a-IGZO TFT-based ISFET sensor platform: (**a**) TG-biasing TG-sensing (TG mode), (**b**) BG-biasing TG-sensing (BG mode), and (**c**) SG-biasing TG-sensing (SG mode).

**Figure 5 sensors-21-04748-f005:**
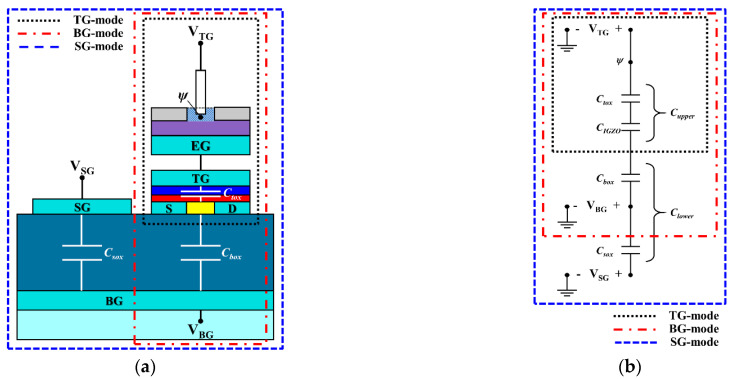
(**a**) Cross-sectional diagram and (**b**) equivalent circuit of a triple-gate a-IGZO TFT-based ISFET sensor platform.

**Figure 6 sensors-21-04748-f006:**
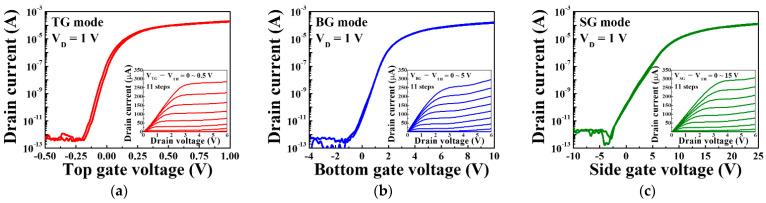
Transfer characteristic curves of triple-gate a-IGZO TFTs operating in the (**a**) TG, (**b**) BG, and (**c**) SG modes. The insets show the output characteristic curves corresponding to each operating mode.

**Figure 7 sensors-21-04748-f007:**
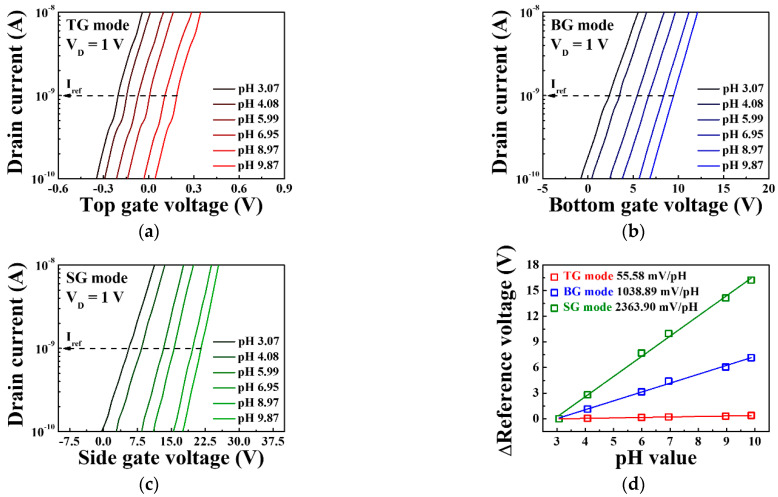
Transfer characteristic curves of the triple-gate a-IGZO TFT-based pH-ISFET sensor platform in (**a**) TG mode, (**b**) BG mode, and (**c**) SG mode. (**d**) Change in the reference voltage of the triple-gate a-IGZO TFT-based pH-ISFET sensor platform as a function of pH.

**Figure 8 sensors-21-04748-f008:**
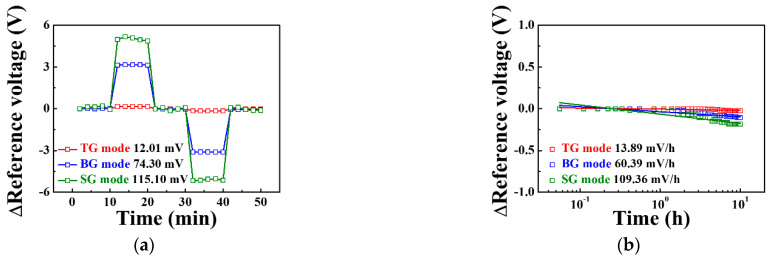
(**a**) Hysteresis voltage and (**b**) drift rate of the triple-gate a-IGZO TFT-based pH-ISFET sensor platform in the TG, BG, and SG modes.

**Figure 9 sensors-21-04748-f009:**
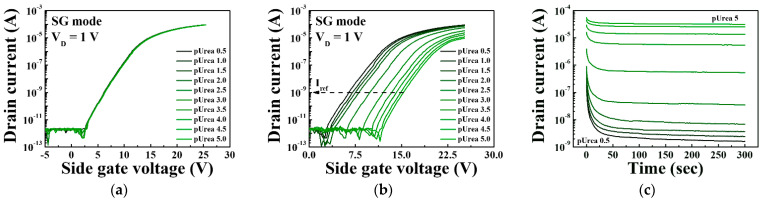
Transfer characteristic curves of the triple-gate a-IGZO TFT-based urea-EnFET sensor platform under the SG mode for (**a**) the interaction between the SnO_2_ membrane and urea and (**b**) the interaction between urease and urea (control test). (**c**) Drain current variation over time for various urea concentrations to confirm the response time needed to reach the equilibrium state of the reaction.

**Figure 10 sensors-21-04748-f010:**
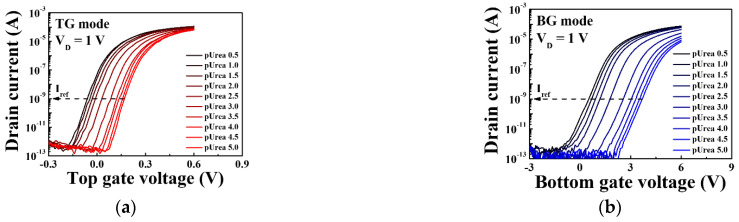
Transfer characteristic curves in urea-urea reactions of the triple-gate a-IGZO TFT-based urea-EnFET sensor platform in the (**a**) TG, (**b**) BG, and (**c**) SG modes for various urea concentrations. (**d**) Change in the reference voltage of the triple-gate a-IGZO TFT-based urea-EnFET sensor platform as a function of pUrea.

**Figure 11 sensors-21-04748-f011:**
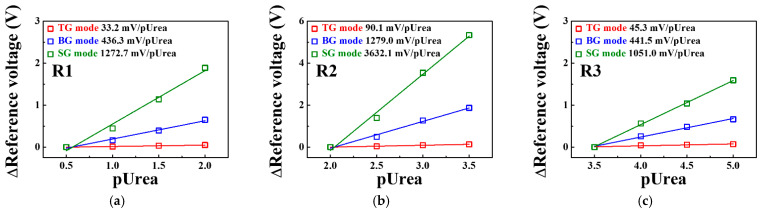
Change in the reference voltage for the TG, BG, and SG modes of the triple-gate a-IGZO TFT-based urea-EnFET sensor platform as a function of pUrea in the (**a**) R1, (**b**) R2, and (**c**) R3 regions.

**Table 1 sensors-21-04748-t001:** Bias conditions for each gate voltage in the three operating modes of the triple-gate a-IGZO TFT-based sensor platform.

	TG Mode	BG Mode	SG Mode
TG	Sweep	Ground	Ground
BG	Ground	Sweep	Floating
SG	NA	NA	Sweep

**Table 2 sensors-21-04748-t002:** Electrical parameters of triple-gate a-IGZO TFTs under top gate, bottom gate, and side gate operations.

Operation Mode	V_TH_ (V)	V_HYS_ (V)	μ_FE_ (cm^2^/V·s)	SS (mV/dec)	I_ON_/I_OFF_ (A/A)
TG	−0.11	0.02	7.37	157.07	6.78 × 10^7^
BG	0.13	0.12	12.89	356.90	5.62 × 10^7^
SG	−0.28	0.27	19.38	434.13	4.48 × 10^7^

**Table 3 sensors-21-04748-t003:** pH sensing parameters of the triple-gate a-IGZO TFT-based pH-ISFET sensor platform in the TG, BG, and SG modes. Amplification refers to the amplified ratio of each mode relative to the TG mode.

Operation Mode	Sensitivity (mV/pH)		Hysteresis Voltage (mV)		Drift Rate (mV/h)	
	Measured	Amplification	Measured	Amplification	Measured	Amplification
TG	55.6	1.0	12.0	1.0	13.9	1.0
BG	1038.9	18.7	74.3	6.2	60.4	4.3
SG	2363.9	42.5	115.1	9.6	109.4	7.9

**Table 4 sensors-21-04748-t004:** Comparison of the analysis characteristics of the proposed urea-EnFET sensor platform and previously reported ISFET-based urea sensors.

Reference	Transducer	Sensitivity (mV/pUrea)	Range (pUrea)	Linearity (%)
[24]	N-channel Si MOSFET fabricated by ITE	61.0	2–3.25	NA
[25]	N-channel Si MOSFET CD4007UB	62.4	2–3.25	98.6
[26]	N-channel Si MOSFET CD4007UBE	43.9	0.5–2	NA
		109.0	2–3.5	NA
		9.8	3.5–5	NA
This work	Triple-gate a-IGZO TFT	1272.7	0.5–2	98.2
		3632.1	2–3.5	99.0
		1051.0	3.5–5	99.8

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
