# Peer review of "Highly Sensitive and Transparent Urea-EnFET Based Point-of-Care Diagnostic Test Sensor with a Triple-Gate a-IGZO TFT"

_sensors, 2021, doi:10.3390/s21144748_

Round 1

Reviewer 1 Report

The authors demonstrate a triple-gate field-effect transistor with an enzymatic extended gate for Urea sensing. Overall, this work provide a great sensitivity when using a triple gate structure. The side gate acts a critical role for high sensitivity. However, it is unclear for the enhancement mechanism of the side gate. Therefore, I suggest to consider the publication of this work after a major revision.

There is a point which may improve the quality of this work.

  1. The side gate seems to show high sensitivity for Urea. However, it is unclear why the sensitivity can be improved so much. The authors should provide more explanation for the side gate enhancement. It will be great if the authors can draw more detailed schematic figures to explain the capacitive coupling in the triple gate system.
  2. The triple gate structure provide such high performance due to the capacitive coupling. However, there is no any capacitance value shown in the manuscript. The authors should provide detailed capacitance estimated from the triple gate system in each mode (TG, BG and SG modes).
  3. The mobility of IGZO in three modes should be discussed in the manuscript.
  4. Current hysteresis should be included in the transfer curves to understand the stability of the electrical performance. 

Author Response

Response Letter to Reviewer #1
We sincerely thank you for giving us such valuable revision suggestions. The reviewers′ comments are really constructive and helpful for us to improve the manuscript. Therefore, we correctly revised the manuscript based on the reviewers′ comments. The following is our point-to-point response to the reviewers′ concerns and descriptions of the revisions, highlighted in yellow in the revised manuscript: 

[Comment 1]
The side gate seems to show high sensitivity for Urea. However, it is unclear why the sensitivity can be improved so much. The authors should provide more explanation for the side gate enhancement. It will be great if the authors can draw more detailed schematic figures to explain the capacitive coupling in the triple gate system.

[Answer 1]
Thank you for your important advice on our manuscript. For the reader's understanding, in Figure 4 of the original manuscript (Figure 5 in the revised manuscript), we have shown a cross-sectional diagram and equivalent circuit for each mode of a triple-gate a-IGZO TFT. In the bottom gate mode, the amplified sensitivity exceeding the Nernst limit is realized through an appropriate combination of Cupper and Clower by capacitive coupling effect. In particular, the amplification ratio increases as the Cupper becomes larger and the Clower becomes smaller. In addition, we introduced a side gate mode to increase the amplification ratio by further reducing the Clower. In the side gate mode, the side gate oxide capacitance and the bottom gate oxide capacitance are connected in series by the bottom floating gate, so that the Clower becomes smaller. Therefore, the side gate mode amplifies the sensitivity much more than the bottom gate mode.

In response to the reviewer’s comments, we have also revised the manuscript in the following section for the reader's understanding:

(Line 221 – Line 233, Page 7 in revised manuscript)
Meanwhile, the proposed triple-gate a-IGZO TFT-based ISFET sensor platform has an excellent SG mode for sensitivity amplification. We introduced the SG mode to increase the amplification ratio by further reducing the Clower. In this case, the Clower of the device is (Cbox×Csox)/(Cbox+Csox) because the floated BG connects Cbox and Csox in series. Csox is the capacitance between the SG and BG and results in the SG gate effectively doubling the thickness of the BG oxide [20]. Therefore, since Clower is smaller in SG mode than in BG mode, SG mode is amplified more than BG mode. For the SG mode using an SG voltage sweep, the capacitive coupling and sensitivity are described in Equation 3 [21]:
ΔV_th^SG=-C_upper/C_lower  Δψ=(C_IGZO C_tox (C_box+C_sox))/((C_IGZO 〖+C〗_tox )(C_box C_sox)) ΔV_th^TG,              (3)
where ΔVthSG is the threshold voltage shift of the SG. In this mode, Cupper is a series combination of Ctox and CIGZO, and Clower is a series combination of Cbox and Csox. In the device proposed in this study, Csox is about 0.015 pF, and the theoretical amplification ratio of SG mode is about 40. This suggests that the sensitivity can be further amplified by using the SG mode of triple-gate a-IGZO TFTs.

[Comment 2]
The triple gate structure provides such high performance due to the capacitive coupling. However, there is no any capacitance value shown in the manuscript. The authors should provide detailed capacitance estimated from the triple gate system in each mode (TG, BG and SG modes).

[Answer 2]
Thank you for your helpful comment. We provided detailed capacitances estimated in the triple-gate system in each mode (TG, BG and SG modes) in the revised manuscript. 

In response to the reviewer’s comments, we have also revised the manuscript in the following section for the reader's understanding:

(Line 213 – Line 233, Page 7 in revised manuscript)
This also implies that TG insulator engineering using high-k materials is effective for increasing Cupper while reducing the leakage current, thereby improving the sensitivity. Nevertheless, increasing the thickness of the BG oxide entails a longer manufacturing process time and higher cost, while thinner TG oxides cause higher leakage currents, making this an inefficient approach. In the device proposed in this study, CIGZO, Ctox, and Cbox are about 1.416 pF, 0.49 pF, and 0.023 pF, respectively, and substituting them into Equation 2, the theoretical amplification ratio of BG mode is about 16.
Meanwhile, the proposed triple-gate a-IGZO TFT-based ISFET sensor platform has an excellent SG mode for sensitivity amplification. We introduced the SG mode to increase the amplification ratio by further reducing the Clower. In this case, the Clower of the device is (Cbox×Csox)/(Cbox+Csox) because the floated BG connects Cbox and Csox in series. Csox is the capacitance between the SG and BG and results in the SG gate effectively doubling the thickness of the BG oxide [20]. Therefore, since Clower is smaller in SG mode than in BG mode, SG mode is amplified more than BG mode. For the SG mode using an SG voltage sweep, the capacitive coupling and sensitivity are described in Equation 3 [21]:
ΔV_th^SG=-C_upper/C_lower  Δψ=(C_IGZO C_tox (C_box+C_sox))/((C_IGZO 〖+C〗_tox )(C_box C_sox)) ΔV_th^TG,              (3)
where ΔVthSG is the threshold voltage shift of the SG. In this mode, Cupper is a series combination of Ctox and CIGZO, and Clower is a series combination of Cbox and Csox. Csox is about 0.015 pF, and the theoretical amplification ratio of SG mode is about 40. This suggests that the sensitivity can be further amplified by using the SG mode of triple-gate a-IGZO TFTs.

[Comment 3]
The mobility of IGZO in three modes should be discussed in the manuscript.

[Answer 3]
Thank you for your proper comment. In order to quantitatively analyze the electrical performance of triple-gate a-IGZO TFTs, we extracted electrical parameters such as threshold voltage, hysteresis voltage, field-effect mobility, subthreshold swing, and on/off current ratio from the transfer characteristic curve.

In response to the reviewer’s comments, we have also revised the manuscript in the following section for the reader's understanding:

(Line 250 – Line 259, Page 8 in revised manuscript)
Table 2 summarizes the electrical parameters extracted from the transfer characteristic curves. A threshold voltage (VTH) of – 0.11 V, hysteresis voltage (VHYS) of 0.02 V, field-effect mobility (μFE) of 7.37 cm2/V·s, subthreshold swing (SS) of 157.07 mV/dec, and on/off current ratio (ION/IOFF) of 6.78 × 107 were obtained under top gate operation. Meanwhile, VTH, VHYS, μFE, SS, and ION/IOFF of 0.13 V, 0.12 V, 12.89 cm2/V·s, 356.90 mV/dec, and 5.62 × 107 were obtained under bottom gate operation, and VTH, VHYS, μFE, SS and ION/IOFF of – 0.28 V, 0.27 V, 19.38 cm2/V·s, 434.13 mV/dec, and 4.48 × 107 were obtained under side gate operation. The TG, BG, and SG sweep operations of the fabricated triple-gate a-IGZO TFT all show excellent electrical characteristics, meaning that they are satisfactory as sensors for sensing pH or urea.

[Comment 4]
Current hysteresis should be included in the transfer curves to understand the stability of the electrical performance.

[Answer 4]
Thank you for your helpful comment. To evaluate the hysteresis characteristic, we measured the transfer characteristic curve in double-sweep mode and extracted the hysteresis voltage. As a result, triple-gate a-IGZO TFT showed very low hysteresis voltage in each mode, showing stable device performance.

In response to the reviewer’s comments, we have also revised the manuscript in the following section for the reader's understanding:

(Line 250 – Line 259, Page 8 in revised manuscript)
Table 2 summarizes the electrical parameters extracted from the transfer characteristic curves. A threshold voltage (VTH) of – 0.11 V, hysteresis voltage (VHYS) of 0.02 V, field-effect mobility (μFE) of 7.37 cm2/V·s, subthreshold swing (SS) of 157.07 mV/dec, and on/off current ratio (ION/IOFF) of 6.78 × 107 were obtained under top gate operation. Meanwhile, VTH, VHYS, μFE, SS, and ION/IOFF of 0.13 V, 0.12 V, 12.89 cm2/V·s, 356.90 mV/dec, and 5.62 × 107 were obtained under bottom gate operation, and VTH, VHYS, μFE, SS and ION/IOFF of – 0.28 V, 0.27 V, 19.38 cm2/V·s, 434.13 mV/dec, and 4.48 × 107 were obtained under side gate operation. The TG, BG, and SG sweep operations of the fabricated triple-gate a-IGZO TFT all show excellent electrical characteristics, meaning that they are satisfactory as sensors for sensing pH or urea.

Again, thank you for your kind consideration and significant advice to our manuscript.

Sincerely yours,
Won-Ju Cho
Department of Electronic Materials Engineering, Kwangwoon University,
20, Gwangun-ro, Nowon-gu, Seoul, 01897, Republic of Korea
E-mail: chowj@kw.ac.kr

Reviewer 2 Report

This paper developed a highly sensitive transparent urea enzymatic field-effect transistor (EnFET) point-of-care (POC) diagnostic test sensor. There are some questions should be answered before accept.

  1. In figure 2, could you provided the SEM photos from different steps, and also EDS analysis could be applied here;
  2. Figure 8 could be moved to the front part, which could help to understand the mechanism of the experiment.

Author Response

Response Letter to Reviewer #2
We sincerely thank you for giving us such valuable revision suggestions. The reviewers′ comments are really constructive and helpful for us to improve the manuscript. Therefore, we correctly revised the manuscript based on the reviewers′ comments. The following is our point-to-point response to the reviewers′ concerns and descriptions of the revisions, highlighted in yellow in the revised manuscript: 

[Comment 1]
In figure 2, could you provided the SEM photos from different steps, and also EDS analysis could be applied here;

[Answer 1]
Thanks for the helpful comments. We provided SEM images to visually demonstrate urease immobilization, and it would be appreciated to note that the SEM images provided are sufficient to verify urease immobilization. As an additional explanation, there was a clear difference in the presence or absence of the urease enzyme in steps (3) and (4). On the other hand, the difference in the surface treatment process from step (1) to step (3) was hardly recognizable by SEM. Therefore, since there was no difference in the images from step (1) to step (3), we provided steps (3) and (4) as representative images to show that the urease enzyme was clearly formed. 

[Comment 2]
Figure 8 could be moved to the front part, which could help to understand the mechanism of the experiment.

[Answer 2]
Thank you for your helpful comment. To explain the sensing mechanism showing the reaction between urease and urea, we moved the Figure to the introduction part (Figure 1 in the revised manuscript). 

In response to the reviewer’s comments, we have also revised the manuscript in the following section for the reader's understanding:

(Line 58 – Line 62, Page 2 in revised manuscript)
Figure 1 shows the sensing mechanism of the urease enzymes immobilized sensing membrane. Since sensing membrane without immobilized urease does not react with urea, the surface potential does not change. On the other hand, since the urease-immobilized sensing membrane reacts with urea and forms a reaction product, the surface potential changes.

Again, thank you for your kind consideration and significant advice to our manuscript.

Sincerely yours,
Won-Ju Cho
Department of Electronic Materials Engineering, Kwangwoon University,
20, Gwangun-ro, Nowon-gu, Seoul, 01897, Republic of Korea
E-mail: chowj@kw.ac.kr

Reviewer 3 Report

This paper shows a proposal of a highly sensitive transparent urea EnFET point-of-care (POC) diagnostic test sensor using a triple-gate a-IGZO thin-film pH ISFET. The results are impressive, and the manuscript is well organized. Thus, I recommend it to be published in Sensors after minor revisions.

  • In the Introduction, it should be mentioned that the sensitive material used is SnO2, as it is described in the abstract, because it is not clear.
  • I would like to see more information about the conditions of the thin-films depositions.
  • Although the authors refer that the two substrates (SnO2 sensing membrane EG and triple-gate a-IGZO TFTs) are electrically connected as shown in Figure 1(d), I think it is not clear how this connection is done.
  • In Figure 9b, I think there is an error in “VD - 1V”.

Author Response

Response Letter to Reviewer #3
We sincerely thank you for giving us such valuable revision suggestions. The reviewers′ comments are really constructive and helpful for us to improve the manuscript. Therefore, we correctly revised the manuscript based on the reviewers′ comments. The following is our point-to-point response to the reviewers′ concerns and descriptions of the revisions, highlighted in yellow in the revised manuscript: 

[Comment 1]
In the Introduction, it should be mentioned that the sensitive material used is SnO2, as it is described in the abstract, because it is not clear.

[Answer 1]
Thank you for the appropriate comment. We revised the Introduction by describing the urease-immobilized SnO2 sensing membrane EG.

In response to the reviewer’s comments, we have also revised the manuscript in the following section for the reader's understanding:

(Line 84 – Line 87, Page 3 in revised manuscript)
In addition, we fabricated a sensing unit by urease immobilization through surface treatment on SnO2 sensing membrane EG. We implemented the urea-EnFET sensor by electrically connecting triple-gate a-IGZO TFTs and urease-immobilized SnO2 sensing membrane EG.

[Comment 2]
I would like to see more information about the conditions of the thin-films depositions.

[Answer 2]
  Thank you for your proper comment. We described the conditions of the thin-film deposition in the revised manuscript.

In response to the reviewer’s comments, we have also revised the manuscript in the following section for the reader's understanding:

(Line 106 – Line 111, Page 3 in revised manuscript)
ITO was deposited at a deposition power of 100 W, a working pressure of 3 mTorr in an ambient of Ar with a flow rate of 20 sccm, and SiO2 was deposited at a deposition power of 200 W, a working pressure of 3 mTorr in an ambient of Ar/O2 with a flow rate of 30/2 sccm. Then, a 20-nm-thick a-IGZO (In:Ga:Zn = 1:1:1) film was deposited by the RF magnetron sputtering system at a deposition power of 100 W, a working pressure of 6 mTorr in an ambient of Ar with a flow rate of 30 sccm as the active region.

(Line 116 – Line 119, Page 3 in revised manuscript)
SiO2 was deposited at a deposition power of 200 W, a working pressure of 3 mTorr in an ambient of Ar/O2 with a flow rate of 30/2 sccm, and Ta2O5 was deposited at a deposition power of 75 W, a working pressure of 3 mTorr in an ambient of Ar with a flow rate of 20 sccm.

(Line 135 – Line 136, Page 4 in revised manuscript)
SnO2 was deposited at a deposition power of 100 W, a working pressure of 3 mTorr in an ambient of Ar with a flow rate of 30 sccm.

[Comment 3]
Although the authors refer that the two substrates (SnO2 sensing membrane EG and triple-gate a-IGZO TFTs) are electrically connected as shown in Figure 1(d), I think it is not clear how this connection is done. 

[Answer 3]
Thank you for your valuable comment. The urease-immobilized SnO2 sensing membrane EG is connected to the dark box through an electrical cable, which is connected to the top gate of the triple-gate a-IGZO TFT through a positioner.

In response to the reviewer’s comments, we have also revised the manuscript in the following section for the reader's understanding:

(Line 150 – Line 155, Page 5 in revised manuscript)
The urease-immobilized SnO2 sensing membrane EG is connected to the dark box through an electrical cable, which is connected to the top gate of the triple-gate a-IGZO TFT through a positioner. Figure 2(a) is a schematic diagram showing the electrical connection between the fabricated urease-immobilized SnO2 sensing membrane EG and triple-gate a-IGZO TFTs, and a photographic image of the fabricated EG is shown in Figure 2(d).

[Comment 4]
In Figure 9b, I think there is an error in “VD - 1V”.

[Answer 4]
There seems to be some misunderstanding by reviewer about the label "VD = 1 V" in Figure 9(b). This means that the transfer curve measurement was performed by applying a voltage of 1 V to the drain. All transfer characteristics presented in this manuscript were measured with VD = 1 V (not “VD - 1V”). Please refer to Figure 6, Figure 7, Figure 9(a) and Figure 10.

Again, thank you for your kind consideration and significant advice to our manuscript.

Sincerely yours,
Won-Ju Cho
Department of Electronic Materials Engineering, Kwangwoon University,
20, Gwangun-ro, Nowon-gu, Seoul, 01897, Republic of Korea
E-mail: chowj@kw.ac.kr

Round 2

Reviewer 1 Report

The authors have fully addressed the issues I mentioned in the last time. Therefore, I suggest to consider the publication as it is.